# Pathological and Therapeutical Implications of Pyroptosis in Psoriasis and Hidradenitis Suppurativa: A Narrative Review

Piotr K. Krajewski [1,*], Maria Tsoukas [2] and Jacek C. Szepietowski [1]

1   Department of Dermatology, Venereology and Allergology, Wroclaw Medical University, Chalubinskiego 1, 50-368 Wroclaw, Poland; jacek.szepietowski@umw.edu.pl
2   Department of Dermatology, University of Illinois at Chicago, Chicago, IL 60607, USA; tsoukasm@uic.edu
*   Correspondence: piotr.krajewski@umw.edu.pl

**Abstract:** This manuscript explores the role of pyroptosis, an inflammatory programmed cell death, in the pathogenesis of two chronic dermatoses, psoriasis and hidradenitis suppurativa (HS). The diseases, though clinically diverse, share common pathogenetic pathways involving the unbalanced interaction between the adaptive and innate immune systems. This review focuses on the molecular changes in psoriatic and HS skin, emphasizing the activation of dendritic cells, secretion of interleukins (IL-17, IL-22, and TNF-$\alpha$), and the involvement of inflammasomes, particularly NLRP3. This manuscript discusses the role of caspases, especially caspase-1, in driving pyroptosis and highlights the family of gasdermins (GSDMs) as key players in the formation of pores leading to cell rupture and the release of proinflammatory signals. This study delves into the potential therapeutic implications of targeting pyroptosis in psoriasis and HS, examining existing medications like biologics and Janus kinase inhibitors. It also reviews the current limitations and challenges in developing therapies that selectively target pyroptosis. Additionally, the manuscript explores the role of pyroptosis in various inflammatory disorders associated with psoriasis and HS, such as inflammatory bowel disease, diabetes mellitus, and cardiovascular disorders. The review concludes by emphasizing the need for further research to fully elucidate the pathomechanisms of these dermatoses and develop effective, targeted therapies.

**Keywords:** hidradenitis suppurativa; psoriasis; inflammatory dermatoses; pyroptosis; inflammasome; NLRP3; inflammation

## 1. Introduction

Inflammatory skin diseases, also called inflammatory dermatoses, are a group of immune-mediated skin diseases with a complex etiology in which both genetic and environmental (i.e., lifestyle) factors play an essential role [1]. Psoriasis and hidradenitis suppurativa (HS) are two of the many diseases that are encompassed by this term [1]. These two disorders, although clinically very diverse, share several pathogenetic pathways, including, in both cases, an unbalanced interaction between the adaptive and innate immune systems [2]. Psoriasis and HS also share potential environmental factors for the development and exacerbation of both diseases, which include smoking, alcohol intake, obesity, metabolic syndrome, and dysbiosis [3–8]. The molecular changes in the psoriatic skin lead to the activation and proliferation of keratinocytes, causing epidermal hyper- and parakeratosis, as well as neutrophilic infiltration of the epidermis [9]. Moreover, the critical element of HS pathogenesis includes the occlusion of the infundibulum of the pilosebaceous unit, its dilation, and rupture, followed by a creation of inflammatory nodules and tunnels with progressive scarring [7]. Interestingly, similarly to psoriasis, the occlusion of the pilosebaceous unit in HS is initiated by the hyperkeratosis and hyperplasia of the keratinocytes localized mainly in the infundibular epithelium [7,9]. The multitude of ongoing studies on the pathomechanisms of psoriasis and HS have led to its meticulous description.

It is known that in both disorders, the activation of the dendritic cells, secretion of IL-23, and subsequent differentiation of T helper lymphocytes (Th) lead to the increased proliferation of Th17, Th1, and Th22 [7,10,11]. These T-cells cause the excessive production of interleukin (IL)-17, IL-22, and tumor necrosis factor-alpha (TNF-$\alpha$) [7,10,11], which, as a result, drives the vicious inflammatory cycle leading to the further development of CD4+ T-cells and production of inflammatory cytokines [7,10,11]. Moreover, in the pathogenesis of HS, additional pathogenetic pathways have been described [7]. Complement activation by pathogen- and damage-associated molecular patterns (PAMPs and DAMPS) leads to inflammasome activation, the production of caspase-1, and release of IL-1$\beta$ and TNF-$\alpha$ [12]. Although many pathogenetic pathways of both diseases have already been described, the exact pathomechanism is still yet to be fully elucidated. Pathogenetic advancements have led to the development of many pathway-specific medications, including biologics and Janus kinase inhibitors (JAKi) [6,12].

The similar pathomechanisms of the two disorders allow for the interchangeable use of psoriatic drugs in HS patients [2]. Nonetheless, even the newest biological drugs, allowing us to almost entirely eradicate skin lesions in psoriatic patients, frequently present incomparably worse responses in patients with HS [13,14]. Therefore, future research targeting possible inflammatory pathways in both diseases is necessary for the development of new, effective therapies. The aim of this review was to evaluate the existence of pyroptosis in the pathogenesis of hidradenitis suppurativa and psoriasis, as well as to assess its potential therapeutic possibilities (Figure 1).

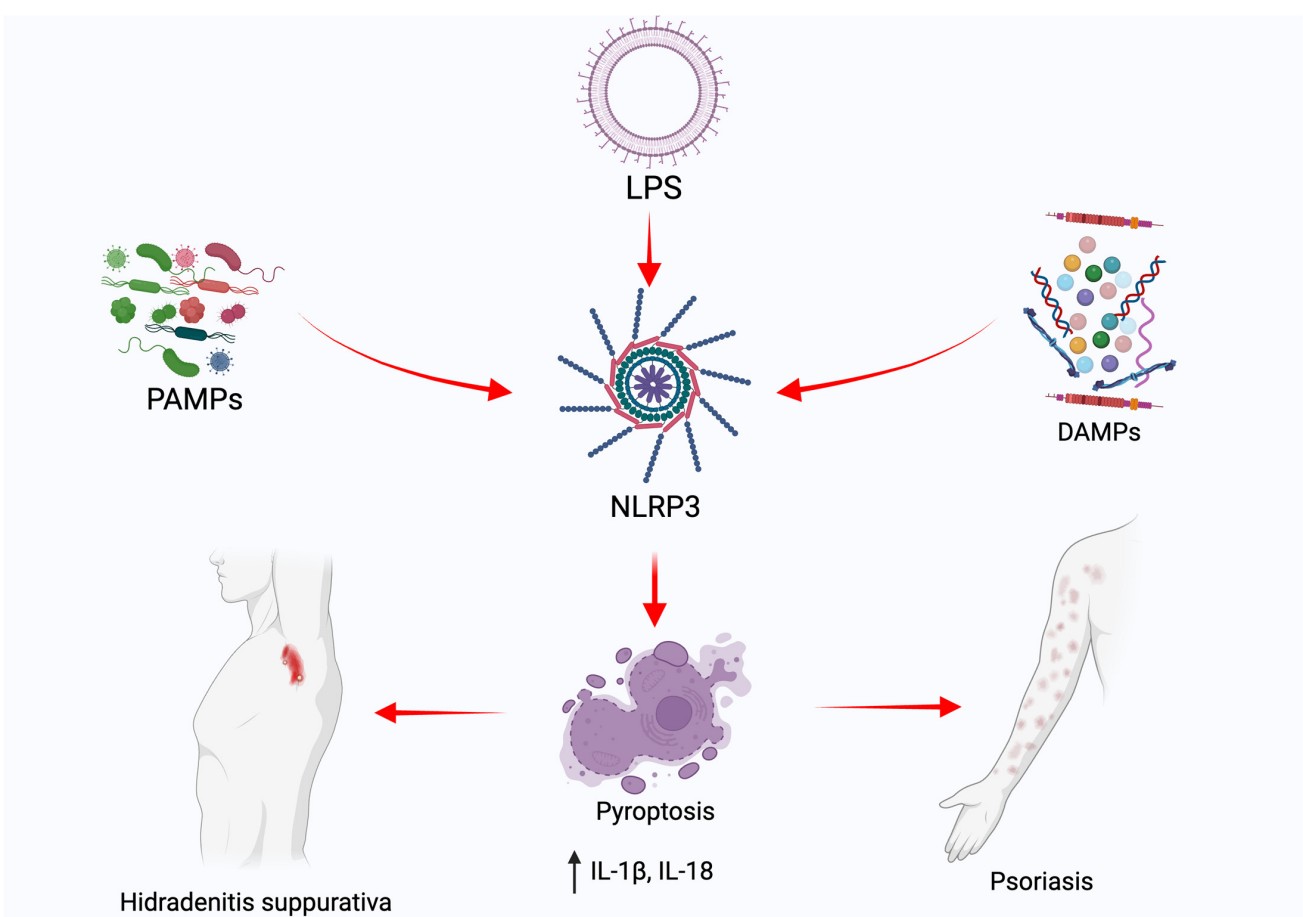

**Figure 1.** Graphical demonstration of the influence of pyroptosis in skin lesions of hidradenitis suppurativa and psoriasis.

## 2. Pyroptosis—Inflammatory, Programmed Cell Death

The removal of damaged and senescent cells is crucial for maintaining the homeostasis of every healthy multicellular organism. There are two basic pathways of programmed cell death, lytic and non-lytic [15]. Initially, only apoptosis was known to be programmed and strictly regulated, while necrosis was considered a form of unregulated cell death [15]. Recently, scientists have discovered differently regulated pathways of necrosis that may be activated by regulated stimuli [16]. Necroptosis and pyroptosis, classified as lytic forms of programmed cell death, were also considered inflammatory due to the release of proinflammatory molecules (DAMPs and cytokines) [16,17]. Both necroptosis and pyroptosis are examples of regulated necrosis; nonetheless, they are triggered by different stimuli and follow different pathways [15]. Pyroptosis, as a form of cell death, was first described in murine macrophage models by Zychlinsky et al. [18] in 1992. At first, it was incorrectly classified as apoptosis and later identified as a form of non-apoptotic cell death [15]. The word pyroptosis comes from Greek and translates into pyro (fire), ptosis (falling) [19]. The pyroptosis begins with the activation of inflammasomes, complex assemblies of multiple proteins activating caspase-1. Various distinct inflammasomes have been identified, each characterized by their unique activators, receptor classes, and the type of activated caspase [20]. The NLR family pyrin domain containing 3 (NLRP3) inflammasome should be underlined due to recent studies on its function in the development of inflammatory dermatoses [21–23].

Inflammasomes are innate immune system protein complexes that act as receptors or sensors to PAMPs and DAMP by promoting the maturation of proinflammatory IL-1β and IL-18 [24]. Inflammasomes form when cellular sensors detect PAMPs or DAMPs. The cellular sensors include, among others, absent in melanoma 2 (AIM2), caspase recruitment domain-containing protein 8 (CARD8), NOD-like receptors (NLRs), and pyrin. These sensors then trigger the activation of the effector enzyme through an adaptor molecule known as ASC (apoptosis-associated speck-like protein containing a CARD) [25]. During pyroptosis, caspase-1 cleaves gasdermin D (GSDMD), releasing its N-terminal domain from the C-terminal domain [26]. Additionally, caspase-11 and its human counterparts, caspase-4 and -5, can trigger pyroptosis by cleaving GSDMD at the same site as caspase-1 [26–28]. Once GSDMD is cleaved, the liberated N-terminal fragments undergo structural changes, leading to their oligomerization in the cellular membrane and the formation of large transmembrane β-barrel pores [29]. These β-barrel pores serve as channels connecting the cytosol to the extracellular space, allowing the release of DAMPs, including mature IL-1β, which attracts immune cells and triggers inflammation [28,30,31]. As previously mentioned, IL-1β has a crucial role in the differentiation and activation of Th17 cells producing IL-17 [32]. Moreover, IL-1β has been reported to be significantly increased in patients with psoriasis and even more in HS, both on the transcriptional and protein levels [33,34]. Furthermore, according to the results of the study performed by our group, HS lesional and non-lesional adjacent skin shows a higher expression of IL-1β in comparison to healthy controls (HC) (data yet not published). While various mechanisms for the secretion of IL-1β from viable cells have been described in recent studies [35], pyroptosis and necroptosis are increasingly recognized as the primary pathways responsible for the release of IL-1β and IL-18, DAMPs, ATP, HMGB1, S100 proteins, and IL-1α [36]. Pyroptosis and apoptosis can be differentiated in several ways [15]. Morphologically, pyroptosis is characterized by a rapid loss of plasma membrane integrity through pore formation, as opposed to the membrane blebbing seen in apoptosis. The subcellular events leading to plasma membrane breakdown in pyroptosis occur within minutes, which is in stark contrast to the relatively slow process of apoptosis. The degradation of membrane structure in pyroptosis leads to the influx of water and ions, resulting in cell swelling and eventual rupture [37]. The non-canonical and canonical pathways are demonstrated in Figure 2.

## Non-canonical and canonical pyroptosis pathways

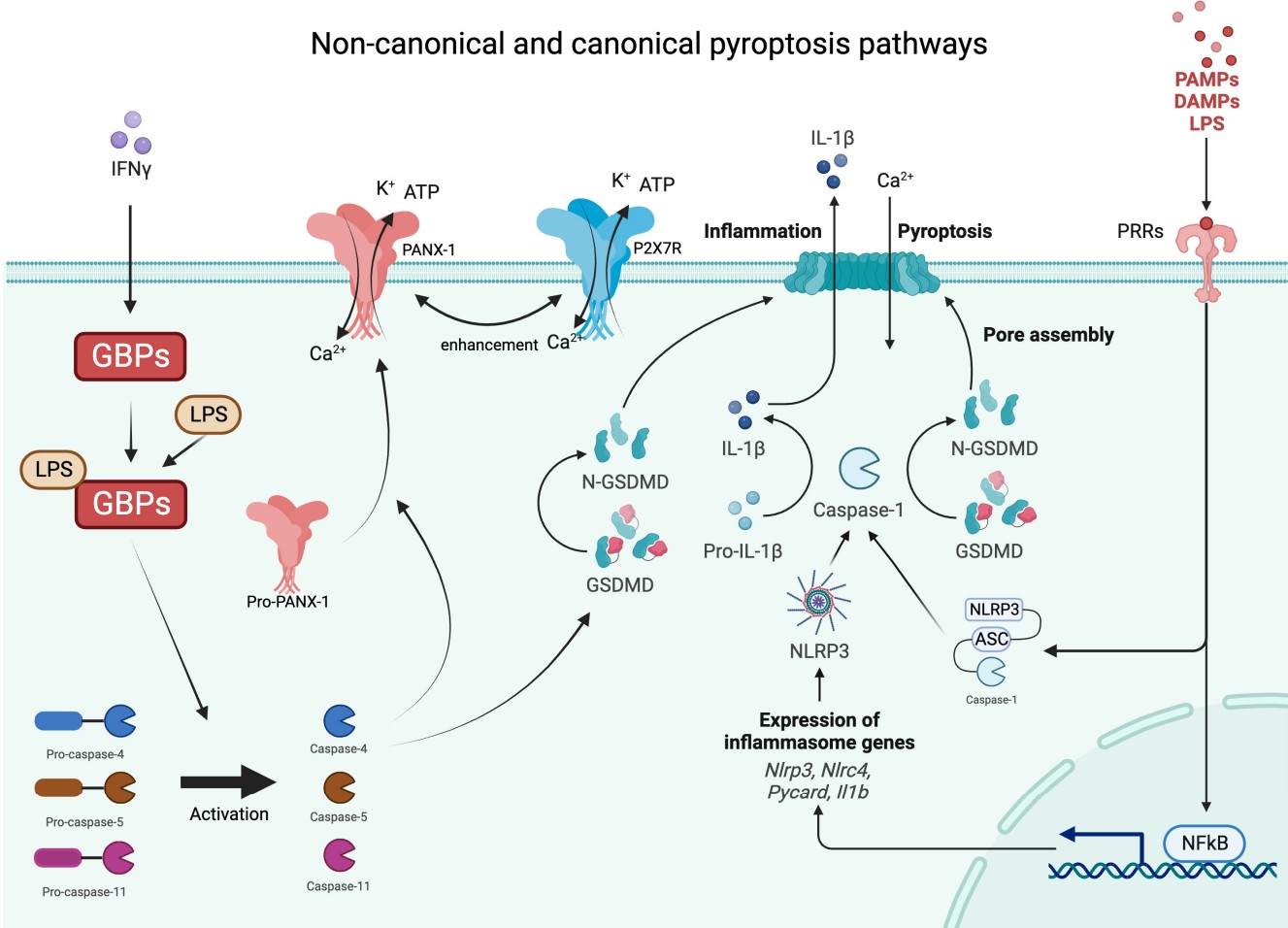

**Figure 2.** Graphical presentations of non-canonical and canonical pathways of pyroptosis. IFNγ—interferon gamma; GBPs—guanylate binding proteins; Ca—calcium; K—potassium; ATP—adenosine triphosphate; PAMPs—pathogen-associated molecular patterns; DAMPs—damage-associated molecular patterns; LPS—lipopolysaccharides; PRRs—pattern recognition receptors; NLRP3—NLR family pyrin domain containing 3 inflammasome; GSDMD—gasdermin D.

### 3. Caspases and Their Role in Pyroptosis

Humans possess four inflammatory caspases (-1, -4, -5, and -11) [15]. The first one to be identified was caspase-1, also called an IL-1-converting enzyme [15]. The caspases are responsible for initiating pyroptosis when inflammasomes are formed [19,24,26,38]. Specifically, caspase-11 and its human counterparts, caspases-4 and -5, function as sensors for cytosolic lipopolysaccharide (LPS), whereas caspase-1 is activated by NLRs and other proteins involved in inflammasome formation [24]. Although caspase-4 and -11 induce pyroptotic cell death by cleaving GSDMD at the same site and with similar efficiency as caspase-1, they do not facilitate the maturation of proinflammatory cytokines IL-1β and IL-18, which are considered hallmarks of pyroptotic cell death in monocyte-derived cells [39]. The division of pro-IL-1β and pro-IL-18 depends on NLRP3 inflammasome activation through caspase-1. Caspase-1-driven GSDMD pores cause membrane damage and K+ efflux, which is known to activate NLRP3. This activation then leads to the processing of IL-1β and IL-18 into their mature forms, as well as the release of other inflammatory cytokines, i.e., TNF-α. These findings underscore GSDMD's dual role in pyroptosis: executing pyroptosis itself and amplifying its effects [15]. Data on the involvement of the caspase-1 pathway in the pathogenesis of HS have been reported. Firstly, Kelly et al. [40] detected the activated caspase 1 in HS biopsies. Moreover, the authors found an association between caspase-1 and increased expression of the NLRP3 inflammasome and IL-18 [40].

Additionally, the inhibition of caspase-1 was associated with a decrease in IL-1β and IL-18 production, confirming the participation of the pyroptotic pathway in the pathogenesis of HS [40]. The results of the study performed by Sanchez et al. [41] in 2019 confirmed the findings from the previous study. The authors found an increased expression of IL-1β, NLRP3, and caspase 1 in the lesional and perilesional skin of HS in comparison to HC. Subsequently, the authors created an ex vivo culture of HS biopsies, which showed an increased concentration of IL-1β after a 4-day culture with no increase in NLRP3 inflammasome or caspase-1 [41]. Similarly, the psoriatic skin demonstrates an increased activity of caspase-1 and elevated levels of caspase-1, -4, and -5 [42,43].

## 4. NLRP3 Inflammasome

The NLRP3 inflammasome is a complex consisting of three basic components: the NLRP3 protein, the adapter protein ASC, and pro-caspase-1 [44]. The NLRP3 protein also consists of three elements, out of which the central nucleotide-binding and oligomerization domain (NACHT) is responsible for oligomerization, the interaction between NLRP3 protein and ASC, as well as for the recruitment of pro-caspase-1 into the protein ASC adapter [45]. The signaling pathway of the NLRP3 inflammasome involves a two-step process: initiation and activation. In the initiation step, PAMPs and/or DAMPs are recognized by pattern recognition receptors (PRRs), such as toll-like receptors, NOD-containing proteins 1 and 2 (NOD1 and NOD2), IL-1 receptor (IL-1R), and TNF receptor 1 and 2 (TNFR1 and TNFR2), leading to the stimulation of cytokine production [46]. NF-κB serves as a critical mediator, acting as the central signal required to prime and activate the NLRP3 inflammasome [47]. Afterward, when the primed cell senses another stimulus, the activation and formation of NLRP3 inflammasome occur, following caspase-1 activation and the release of IL-1β and IL-18 [47]. It has been previously reported that NRLP3 may be associated with a variety of diseases, including inflammatory bowel diseases, atopic dermatitis, atherosclerosis, and diabetes mellitus type 2 [48–50]. The association of the NRLP3 pathway with HS was also reported. Authors found overexpression of the NRLP3 inflammasome, as well as caspase-1, in the skin of HS patients [23]. This was also confirmed on the protein level. Western blot analyses showed an increased presence of active forms of NLRP3-associated proteins, including caspase-1, in the keratinocytes of HS-affected skin [21]. Moreover, increased expression of the NRLP3 gene was found in HS lesional (19.4-fold) and perilesional (6.3-fold) skin in comparison to HC (study performed by our group, with data not yet published). Moreover, the relative mRNA levels were significantly higher in the skin of HS patients than in HCs (data not yet published). It was also shown that the inhibition of NLRP3 leads to a significant decrease in associated inflammatory proteins, e.g., IL-1β, IL-17A, TNF-α, and IL-18 [23]. The expression of NLRP3 and associated inflammatory cytokines was also increased in psoriatic skin in comparison to HC [51]. Similar observations were made for imiquimod-induced skin lesions in a mouse model [52]. Moreover, it was reported that quinolinic acid, ginsenoside, 3H-1,2-Dithiole-3-Thione, curcumin, and bortezomib might inhibit NLRP3 inflammasome activation [53–56]. Alrefai et al. [57] described an association of NLRP3 (rs10754558) gene polymorphism with psoriasis.

## 5. The Family of Gasdermins and Their Role in Pyroptosis

Gasdermins (GSDMs) are a family of six proteins identified in 2007 whose expression was first described in the gastrointestinal tract and skin [58]. Their biological functions were not known for more than a decade. However, recent studies show that GSDMs, and especially GSDM D, are a key component in the pyroptotic pathway [28]. The mechanism behind the activation of GSDMD was simultaneously studied by Kayagaki et al. and Shi et al. [26,28]. Both groups described genes associated with GSDMD, which, when mutated or deleted, would prevent LPS-induced cell death [26,28]. The GSDMD consists of two domains, a pore-forming N-terminal domain (GSDMD-N) and an inhibitory C-terminal domain (GSDMD-C), which are separated by an unorganized linker region [26,28]. The

activation of GSDMD may occur, as previously mentioned, in one of two ways: a canonical one, through NLRP3 inflammasome and caspase-1; or non-canonical, through LPS and caspase-11 [59]. During the cleavage, SGDMD-Ns relocate to the plasma, binding acidic lipids, which include phosphatidylserine (PS) or cardiolipin (CL), and oligomerizing into a ring-like structure, which is subsequently inserted into the cellular membrane to form a large pore [60]. The diameter of the pore enables the passage of multiple soluble cytosolic molecules, including IL-1β and IL-18 [61]. Afterward, the flow through the pore causes the disruption of osmotic potential, cell membrane rupture, and pyroptotic death [62]. Interestingly, similar mechanisms have been observed in bacterial membranes, which proves that GSDMD may also play a role in pathogen destruction [63].

Besides GSDMD, several other GSDMs possess pore-forming activity, including GSDMA3 and GSDME [60]. Wang et al. [64] observed in 2017 that GSDME, once activated by caspase-3, creates pores, facilitating pyroptosis. It is noteworthy that caspase-3 is conventionally known as an apoptotic caspase, highlighting GSDME's role in connecting various cell death pathways. Furthermore, GSDME acts as a tumor suppressor by enhancing pyroptosis triggered by chemotherapeutic agents that activate caspase-3. It is often silenced in numerous tumor types. In cases where cancer cells express elevated levels of GSDME, they undergo pyroptosis in response to chemotherapies that typically induce apoptosis, as shown in multiple studies [65,66]. Further investigation into the control of GSDME expression and activation will provide insights into the intricate interplay among multiple cell death pathways.

Several studies were performed assessing the possible involvement of GSDMs in the pathogenesis of psoriasis. Nowowiejska et al. [67] found elevated serum levels of GSDMD in patients suffering from psoriasis, as well as increased expression of GSDMD in psoriatic skin in comparison to non-lesional and healthy skin. No correlations with psoriasis severity were found [67]. Likewise, Lian et al. [68] found aberrantly expressed GSDMD in the dermis of psoriatic plaques. Moreover, the application of disulfiram (inhibitor of pyroptosis) ameliorated imiquimod-induced psoriasis-like dermatitis [68]. Nowowiejska et al. [69] also found that serum concentrations of GSDME were significantly higher in psoriatic patients in comparison to HC. As for GSDMD, the expression of GSDME was significantly increased in psoriatic plaques; however, no correlations with psoriasis severity were found [69]. Interestingly, the authors found statistically significant correlations between GSDME serum levels and ALT (r = $-0.386$, $p = 0.005$), triglycerides (r = $-0.324$, $p = 0.003$), serum glucose (r = $-0.435$, $p = 0.002$), uric acid (r = $-0.471$, $p = 0.006$), creatinine (r = $-0.283$, $p = 0.04$), and glomerular filtration rate (r = $0.292$, $p = 0.04$) [69]. The authors hypothesized a protective role of GSDME in metabolic disorders associated with psoriasis [69]. In 2023, Ji et al. [70] reported a statistically significant GSDMB on both transcriptional and protein levels in patients with psoriasis vulgaris in comparison to HC. Furthermore, the downregulation of the GSDMB gene resulted in decreased keratinocyte proliferation, an inhibition of pyroptosis, and an increase in apoptosis [70]. The authors hypothesized that GSDMB could play a role in keratinocyte differentiation and therefore in the pathogenesis of psoriasis [70].

To the best of our knowledge, there were no studies on the possible implication of GSDMs in the pathogenesis of HS.

## 6. The Role of Pyroptosis in Healthy Organisms

Recent findings suggest that pyroptosis may serve as an efficient innate defense mechanism against infections in the host [24]. To elaborate, pathogens can trigger inflammasomes (mostly NLRP3), leading to pyroptosis and the subsequent destruction of infected cells, thereby exposing the pathogens to extracellular immune defenses. This exposure can facilitate the recruitment of neutrophils, bolstering the host's defense against infection [24]. Additionally, during pyroptosis, inflammatory cytokines like IL-1β and IL-18 are secreted. IL-1β not only incites vasodilation, inflammation, and immune cell infiltration but also plays a role in shaping adaptive immune responses [71]. IL-18, on the other hand, promotes interferon (IFN)-γ production in various immune cells (including Th1 and natural

killers [NK]) and influences the development of Th2-specific immune responses [72]. These signaling molecules also attract immune cells to the infection site, aiding in pathogen clearance [72]. Furthermore, pyroptosis leads to the release of extracellular inflammatory agents, including IL-1, heat-shock proteins, and ATP, which activate pattern recognition receptors (PRRs), triggering the production of proinflammatory cytokines. This helps the host to manage and eliminate microbial infections, ultimately restoring tissue homeostasis [73]. Nonetheless, various studies examining pyroptosis have uncovered its dual role as both a contributor to and a remedy for certain issues. In addition, moderate pyroptosis proves beneficial as it efficiently eliminates infected cells, preserving cellular equilibrium and effectively curbing excessive cell proliferation. Moreover, the release of inflammatory signals during pyroptosis, often referred to as "to die for life" signals, heightens the immune response to combat ongoing infections. Conversely, excessive pyroptosis can exacerbate inflammatory symptoms, leading to cell demise and severe tissue and organ dysfunction. These processes are linked to the pathogenesis of specific diseases and disruptions in in vivo homeostasis [15,25,39,46,73]. The current literature underlines the activation of pyroptosis in various inflammatory and neoplastic diseases [73]. It has been reported that pyroptosis plays a role in the development, growth, and metastases of melanoma, breast, colorectal, gastric, lung, cervical cancer, hepatocellular carcinoma, and leukemia [73]. Pyroptosis is also associated with various inflammatory disorders, including atherosclerosis and a subsequent myocardial infarction, diabetic cardiomyopathy, and hypertension; neurological disorders, including Parkinson's and Alzheimer's diseases, stroke, and amyotrophic lateral sclerosis; metabolic diseases such as diabetes, obesity, and gout; as well as autoinflammatory disorders like inflammatory bowel disease [15,25,39,46,73].

## 7. The Role of Pyroptosis in Inflammatory Disorders Associated with Psoriasis/HS

### 7.1. Inflammatory Bowel Disease (IBD)

The coexistence of IBDs and chronic inflammatory skin disorder has been frequently reported [74]. In a cohort study performed by Schneeweis et al. [74], the authors found that patients with HS are at a higher risk of developing ulcerative colitis (UC) and Crohn's disease (CD) (Hazard Ratio [HR] of 2.3 and 2.7, respectively). In contrast, psoriatic patients suffer 1.2 times more frequently from CD, but not UC, than HC [74]. Shared inflammatory pathways, including TNF, IL-17, and IL-23, are key inflammatory elements in the development of IBD [75]. Moreover, clinical trials have shown that similar therapeutic factors may alleviate symptoms of all the above-mentioned diseases [76–78]. Pyroptosis may also play a role in developing UC and CD gastrointestinal lesions. It controls bacterial and viral infections of the intestinal epithelium and stimulates host defenses from intracellular pathogens by enclosing them in pore-induced intracellular traps (PITs) in macrophages [79]. PITs additionally promote the innate immune response, which, as a result, recruits more inflammatory cells, including neutrophils and macrophages [79]. The role of GSDMs in the development of IBDs is complex and still remains unclear. Patients who suffer from IBD present higher expressions of GSDMD and GSDME in the intestinal epithelium [80–82]. Furthermore, mRNA expressions of Nima associated kinase 7 (NEK7), a protein interacting with NRLP3, Caspase-1, and GSDMD, were significantly higher in UC patients in comparison to HC [83,84]. Likewise, NEK7 deficiency was associated with a lower expression of NRLP3 itself, Caspase-1, and GSDMD [83]. Moreover, GSDMD deficiency aggravates experimental colitis through cyclic GMP-AMP synthase (cGAS)-dependent inflammation [85]. On the other hand, the inflammation caused by GSDME was associated with the development of colorectal cancer associated with chronic colitis [80].

### 7.2. Diabetes Mellitus (DM)

Obesity, metabolic syndrome, and diabetes are some of the most commonly mentioned comorbidities of psoriasis and HS. The pathogenesis of those comorbidities remains unclear. However, mutual inflammatory pathways, shared cellular mediators, genetic susceptibility, and risk factors of these diseases have been described [86,87]. The role of pyroptosis in the

pathophysiology of diabetes and diabetic neuropathy has been vastly studied. Increased expressions of NLRP3 and IL-1β were observed in the non-obese diabetic mice (type 1 DM) [88]. Conversely, saturated fatty acids, which promote the development of type 2 DM activated NLRP3, promote the production of IL-1β and the further activation of caspase-1 in macrophages [89]. On the contrary, unsaturated fatty acids inhibited the activation of NLRP3, resulting in decreased activation of caspase-1, thus preventing the development of obesity and DM2 [90]. It has been recently revealed that thioredoxin-interacting protein (TXNIP), a key element of diabetes progression, activates the NLRP3 inflammasome and subsequent pancreatic B-cell pyroptosis in diabetic mice [91]. It was also reported that the inhibition of TXNIP is effective in preventing diabetes complications [92]. In individuals with diabetes, there was a statistically significant increase in the levels of NLRP3, ASC, and subsequent proinflammatory factors [93]. Conversely, a reduction in calorie intake and weight loss through exercise among obese individuals with type 2 diabetes resulted in a significant decrease in mRNA levels of NLRP3 and IL-1β. This intervention also led to an improvement in insulin sensitivity [94]. In summary, pyroptosis plays a role in inflammation, the destruction of beta cells, insulin resistance, and the development of diabetes.

### 7.3. Cardiovascular Disorders

Patients suffering from psoriasis and HS are at a higher risk of developing life-threatening cardiovascular diseases (CVDs) due to a common prevalence of CV risk factors, as well as a common inflammatory pathway, which leads to the creation of atherosclerosis [87,95]. Research has indicated that pyroptosis plays a significant role in the development of various cardiovascular diseases. The occurrence of pyroptosis in endothelial cells, macrophages, and smooth muscle cells is closely linked to the initiation and progression of atherosclerotic lesions, impacting the stability of arterial plaques [96–99]. Additionally, the presence of NLRP3/IL-1β/caspase-1 has been detected in individuals with conditions such as diabetes, myocardial infarction, arrhythmia, and cardiac hypertrophy. Some treatments have shown promise in alleviating cardiovascular disease symptoms by reducing pyroptosis [100]. Among others, colchicine, a non-specific inhibitor of NLRP3, has been shown to significantly reduce infarct size in phase II clinical trials. Moreover, the inhibition of caspase-1, IL-1, and IL-18 may restore heart function [100]. In recent decades, an increasing body of evidence has pointed towards atherosclerosis as being an inflammatory disease linked to impaired endothelial function, with pyroptosis emerging as a critical contributor to this phenomenon [97,101,102]. Notably, components of the NLRP3 inflammasome, including NLRP3, ASC, and caspase-1, exhibit elevated expression within carotid atherosclerotic plaques, implying their involvement in atherosclerosis pathogenesis [103,104]. The adverse impact of NLRP3 on atherosclerosis primarily hinges on the action of its downstream cytokine IL-1β. Studies involving IL-1β-deficient mice reveal a nearly 30% reduction in atherosclerotic plaque size, possibly indicative of impaired monocyte migration to lipid deposition sites [105,106]. These findings affirm the role of pyroptosis in atherosclerotic plaque formation and suggest that targeting NLRP3 and IL-1β within the pyroptosis pathway could represent a potential strategy for atherosclerosis treatment [100].

## 8. Potential Therapeutic Target

The use of drugs to block the activation of NLRP3, and therefore pyroptosis, has shown significant therapeutic benefits in diverse rodent models of inflammatory and cancerous conditions [107–109]. These outcomes align with the results obtained when NLRP3 is genetically removed, emphasizing the potential of NLRP3 as a viable target for pharmaceutical intervention [110,111]. Nevertheless, it is essential to underline that the majority of studies on pyroptosis as a potential therapeutic target are still in the primary research stage. Currently, the only available option for inhibiting NLRP3-mediated effectors in clinical practice in both HS and psoriasis is drugs targeting IL-1β, like anakinra and canakinumab. Anakinra, an IL-1R antagonist, has been previously studied in generalized and palmo-

plantar pustular psoriasis [112–115]. The results of these studies are contraindicatory, and therefore its efficacy is impossible to assess [112–115]. Moreover, it has been reported that anakinra itself may produce the onset of psoriasis and psoriatic-like lesions [116,117]. Similarly, canakinumab, a monoclonal antibody targeting IL-1β, according to available publications has been only used in the treatment of pustular psoriasis [118]. However, both anakinra and canakinumab have been used for the treatment of HS, with ambiguous results. According to a randomized clinical trial performed by Tzanetakou et al. [119], 78% (seven out of nine) patients achieved clinical response after 12 weeks of treatment. Moreover, patients taking anakinra would have a prolonged time to a new HS exacerbation [119]. Furthermore, several case series and case reports have reported successful treatment of HS with anakinra [120–123]. Yet, other authors indicate the failure of anakinra treatment in HS patients [124,125]. No randomized clinical trial has been performed yet for canakinumab in HS, but case results are contraindicatory. According to some authors, canakinumab may offer rapid and at least partial response [126–128], while others indicate treatment failure [129,130]. Rilonacept, also known as IL-1 Trap, represents a third clinically approved therapy targeting IL-1. This human dimeric fusion protein is composed of the extracellular domains of IL-1R1 and IL-1RAcP fused to the Fc fragment of IgG. Acting as a soluble decoy receptor, it hinders the binding of both IL-1α and IL-1β to IL-1R1. Notably, it has demonstrated efficacy in managing symptoms in a cohort of patients with Schnitzler's syndrome. However, no studies on psoriasis or HS were performed [25]. Taking into consideration all of the above-mentioned studies, one may conclude that, currently, there is no treatment selectively targeting pyroptosis and NLRP3 inflammasome activation in psoriasis and HS.

## 9. Conclusions

In conclusion, in this review, we have clearly indicated that inflammation-related pyrosis constitutes an inseparable element of the pathogenesis of chronic dermatoses, including psoriasis and hidradenitis suppurativa. Moreover, it has been shown that pyroptosis also plays a role in the development and exacerbation of comorbidities occurring in patients suffering from psoriasis and HS. The currently ongoing research, which mostly reports increased expressions of pyroptosis-related cytokines and proteins, opens the possibility of future research on highly effective and selective therapeutic options. Unfortunately, the current possibilities of blocking pyroptosis and NLRP3 activation in psoriasis and HS are very limited and show ambiguous efficacy.

**Author Contributions:** Conceptualization, M.T., J.C.S. and P.K.K.; methodology, M.T., J.C.S. and P.K.K.; investigation, P.K.K.; data curation, P.K.K.; writing—original draft preparation, M.T., J.C.S. and P.K.K.; writing—review and editing, M.T., J.C.S. and P.K.K.; visualization, P.K.K.; supervision, M.T. and J.C.S.; project administration, M.T. and J.C.S.; funding acquisition, P.K.K. and J.C.S. All authors have read and agreed to the published version of the manuscript.

**Funding:** This research was funded by Wroclaw Medical University, research grant number SUBK.C260.23.037.

**Acknowledgments:** Figure adapted from "Suppression of Inflammasome by IRF4 and IRF8 is Critical for T Cell Priming", by BioRender.com (2023). Retrieved from https://app.biorender.com/biorender-templates (accessed on 7 January 2024).

**Conflicts of Interest:** The authors declare no conflicts of interest.

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
