# Peer review of "Pathological and Therapeutical Implications of Pyroptosis in Psoriasis and Hidradenitis Suppurativa: A Narrative Review"

_cimb, doi:10.3390/cimb46010043_

Round 1

Reviewer 1 Report

Comments and Suggestions for Authors

Thank you for the opportunity to read the text entitled: Pathological and therapeutic implications of pyroptosis in chronic inflammatory dermatoses: a narrative review. The work submitted for review is a short, synthetic presentation of the current state of knowledge about the mechanism and role of pyroptosis in the pathomechanism of two selected dermatological diseases. The text is well written. It is valuable and worthy of publication in the journal: Current Issues in Molecular Biology. It also completely agrees with the topic of the special issue to which it was submitted.

However, I would like to ask you to consider a few corrections which, in my opinion, will improve the quality of the work: 

1. Please change and clarify the title of the paper, clearly indicating which two diseases will be discussed in this text.

2. Please think about the layout of the manuscript (division into chapters). It is worth following the guidelines of the journal or publisher in this regard.

3. Please provide a broader discussion of two aspects raised in the text: lines 254-255 and 297-298.

4. The list of references is extensive and correctly selected. However, please provide the DOI number for items that have this index.

Author Response

Dear Reviewer:

Thank you very much for the review and for your time. We have revised our manuscript according to your comments. All the introduced changes are highlighted in the manuscript.

  1. Thank you for the opportunity to read the text entitled: Pathological and therapeutic implications of pyroptosis in chronic inflammatory dermatoses: a narrative review. The work submitted for review is a short, synthetic presentation of the current state of knowledge about the mechanism and role of pyroptosis in the pathomechanism of two selected dermatological diseases. The text is well written. It is valuable and worthy of publication in the journal: Current Issues in Molecular Biology. It also completely agrees with the topic of the special issue to which it was submitted.

However, I would like to ask you to consider a few corrections which, in my opinion, will improve the quality of the work:

  • Thank you very much for this comment. We are very glad that you consider our paper valuable

  1. Please change and clarify the title of the paper, clearly indicating which two diseases will be discussed in this text.

  • We acknowledge your comment. The title was changed into: “Pathological and therapeutical implications of pyroptosis in psoriasis and hidradenitis suppurativa: a narrative review.”

  1. Please think about the layout of the manuscript (division into chapters). It is worth following the guidelines of the journal or publisher in this regard.

  • Thank you for your comment. The CIMB layout or even MDPI layout is more for systematic reviews or original articles. It follows the traditional introduction, methods, results, and discussion, which in this article would not be possible. The current layout allows readers to go into the part that interests them directly.

  1. Please provide a broader discussion of two aspects raised in the text: lines 254-255 and 297-298.

  • We acknowledge your comment. The following paragraphs were added:
  • “The current literature underlines the activation of pyroptosis in various inflammatory and neo-plastic diseases [73]. It has been reported that pyroptosis plays a role in the development, growth, and metastases of melanoma, breast, colorectal, gastric, lung, cervical cancer, hepatocellular car-cinoma and leukemia [73]. Pyroptosis is also associated with various inflammatory disorders, including atherosclerosis and a subsequent myocardial infarction, diabetic cardiomyopathy, and hypertension; neurological disorders, including Parkinson’s and Alzheimer’s diseases, stroke, and amyotrophic lateral sclerosis; metabolic diseases with diabetes, obesity, and gout; as well as autoinflammatory disorders like inflammatory bowel disease [15, 25, 39, 46, 73].”
  • Among others, colchicine, a non-specific inhibitor of NLRP3, has been shown to reduce infarct size in phase II clinical trials significantly. Moreover, inhibition of caspase-1, IL-1 and IL-18 may re-store heart functions [100].

  1. The list of references is extensive and correctly selected. However, please provide the DOI number for items that have this index.

  • We acknowledge your comment. The references were composed according to the instruction for authors – the CIMB reference format does not include DOI numbers.

Reviewer 2 Report

Comments and Suggestions for Authors

Dear authors,

This is an excellent paper that present a complex analysis about pyroptosis in psoriasis and HS. I suggest to add information’s about therapeutic target (biologics) that are used now to treat psoriasis and HS.

Author Response

Dear Reviewer:

Thank you very much for the review and for your time. We have revised our manuscript according to your comments. All the introduced changes are highlighted in the manuscript.

  1. This is an excellent paper that present a complex analysis about pyroptosis in psoriasis and HS. I suggest to add information’s about therapeutic target (biologics) that are used now to treat psoriasis and HS.

  • Thank you very much for this comment. We would like to underline that the only biologics that are currently used in psoriasis and HS and could block pyroptosis are anakinra and canakinumab. Both are described in the 8th paragraph entitled “Potential therapeutic target”
  • We have added a little information about an additional IL-1 blocker – rilonacept “Rilonacept, also known as IL-1 Trap, represents a third clinically approved therapy targeting IL-1. This human dimeric fusion protein is composed of the extracellular domains of IL-1R1 and IL-1RAcP fused to the Fc fragment of IgG. Acting as a soluble decoy receptor, it hinders the bind-ing of both IL-1α and IL-1β to IL-1R1. Notably, it has demonstrated efficacy in managing symp-toms in a cohort of patients with Schnitzler's syndrome, however no studies on psoriasis or HS were performed [25]”

Reviewer 3 Report

Comments and Suggestions for Authors

What is the relationship between the treatment of psoriasis with biologics and inflammation-related pyrosis?

Author Response

Dear Reviewer:

Thank you very much for the review and for your time. We have revised our manuscript according to your comments. All the introduced changes are highlighted in the manuscript.

  1. What is the relationship between the treatment of psoriasis with biologics and inflammation-related pyrosis?

  • Thank you very much for this comment. We would like to underline that the only biologics that are currently used in psoriasis and HS and could block pyroptosis are anakinra and canakinumab. Both are described in the 8th paragraph entitled “Potential therapeutic target”
  • We have added a little information about an additional IL-1 blocker – rilonacept “Rilonacept, also known as IL-1 Trap, represents a third clinically approved therapy targeting IL-1. This human dimeric fusion protein is composed of the extracellular domains of IL-1R1 and IL-1RAcP fused to the Fc fragment of IgG. Acting as a soluble decoy receptor, it hinders the bind-ing of both IL-1α and IL-1β to IL-1R1. Notably, it has demonstrated efficacy in managing symp-toms in a cohort of patients with Schnitzler's syndrome, however no studies on psoriasis or HS were performed [25]”

Round 2

Reviewer 2 Report

Comments and Suggestions for Authors

Dear authors,

Congratulation for your work.